# Digitally Enhanced Methods for the Diagnosis and Monitoring of Treatment Responses in Actinic Keratoses: A New Avenue in Personalized Skin Care

**DOI:** 10.3390/cancers16030484

**Published:** 2024-01-23

**Authors:** Cristina Soare, Elena Codruta Cozma, Ana Maria Celarel, Ana Maria Rosca, Mihai Lupu, Vlad Mihai Voiculescu

**Affiliations:** 1Department of Oncological Dermatology, Carol Davila University of Medicine and Pharmacy, 050474 Bucharest, Romania; cristina.vajaitu@gmail.com (C.S.); lupu.g.mihai@gmai.com (M.L.); voiculescuvlad@yahoo.com (V.M.V.); 2Department of Pathophysiology, University of Medicine and Pharmacy of Craiova, 200349 Craiova, Romania; 3Department of Dermatology and Allergology, Elias University Emergency Hospital, 011461 Bucharest, Romania; ana.malciu@gmail.com; 4Department of Dermatology, University Military Hospital “Dr. Carol Davila”, 010825 Bucharest, Romania; anaa.rosca@gmail.com

**Keywords:** actinic keratosis, reflectance confocal microscopy, optical coherence tomography, dermoscopy, skin imaging, dermoscopy guided high frequency ultrasonography

## Abstract

**Simple Summary:**

Actinic keratoses are lesions with a variable potential for malignant transformation, but with a difficult predictability of evolution towards this type of lesions. Non-invasive skin imaging methods represent diagnostic tools that allow a better evaluation of actinic keratoses lesions, as well as the establishment of a diagnosis with greater specificity and sensitivity. This review aims to identify all skin imaging methods that are useful in the diagnosis and monitoring of the treatment of actinic keratoses, as well as to identify those lesions at risk of transforming into squamous cell carcinoma.

**Abstract:**

Non-melanocytic skin cancers represent an important public health problem due to the increasing incidence and the important local destructive potential. Thus, the early diagnosis and treatment of precancerous lesions (actinic keratoses) is a priority for the dermatologist. In recent years, non-invasive skin imaging methods have seen an important development, moving from simple observational methods used in clinical research, to true diagnostic and treatment methods that make the dermatologist’s life easier. Given the frequency of these precancerous lesions, their location on photo-exposed areas, as well as the long treatment periods, with variable, imprecise end-points, the need to use non-invasive imaging devices is increasingly evident to complete the clinical observations in the diagnosis and treatment of these lesions, with the aim of increasing accuracy and decreasing the adverse effects due to long treatment duration. This is the first review that brings together all skin imaging methods (dermoscopy, reflectance confocal microscopy, ultrasonography, dermoscopy-guided high frequency ultrasonography, and optical coherence tomography) used in the evaluation of actinic keratoses and their response to different treatment regimens.

## 1. Introduction

Actinic keratosis (AK), also known as solar keratosis, is a keratinocyte-derived precancerous lesion frequently found in adults, that typically develops on parts of the skin exposed to the sun [1,2,3,4]. Field cancerization, as delineated by a panel of European dermatologists, refers to the anatomical region encompassing or neighboring AK lesions within visibly photodamaged skin. This region is marked by pigmentary alterations, coarse texture, thinning, or telangiectasia, and is where AK lesions evolve into cutaneous Squamous Cell Carcinoma (cSCC) [5]. The primary risk factors for AK and FIC development include long-term ultraviolet radiation exposure, sunbed usage, being over 60 years of age (with up to 80% prevalence in this group), male gender, light skin types (Fitzpatrick I-II), extended periods of immunosuppression, and a past occurrence of AK or non-melanoma skin cancer [6]. Also, certain medications that increase photosensitivity, such as hydrochlorothiazide, have been associated with a greater risk of developing AK [7]. Moreover, all the factors involved in the apparition of those lesions are not identified, AK and FC, in particular, being considered multifactorial, complex diseases. Recently, the hypothesis of the involvement of Human Polyoma Virus (HPyV) in the development of this precancerous lesion has emerged. However, a recent study published by Prezioso et al., who evaluated the prevalence of multiple strains of this virus in AK lesions, could not find a higher prevalence of HPyV in these precancerous lesions [8]. Consistent sunscreen application and sun exposure education represent factors that reduce the risk of AK development [9]. Incidence rates of AK are notably higher in populations subjected to more intense solar UV radiation, such as those in Australia compared to the United Kingdom, with a marked increase in prevalence seen in individuals over 60 years of age compared to those under 40 years [10]. European AK prevalence fluctuates between 4.7% in France to 31% among Austrians over 30 (as determined by dermatologists in clinics) and up to 37% according to a large cohort study [10,11,12]. Although direct comparisons of AK prevalence are challenging due to varying research methods, there is a consistent finding that prevalence is greater in males than females and that it escalates with age.

### Background and Motivation

The need for early diagnosis and treatment of AK is more and more important as a consequence of continuously increasing economic impact as the global population ages and accumulates lifetime sun exposure [12,13,14].

Their localization on photo-exposed areas (head and neck), as well as the presence of multiple lesions of different stages (including subclinical ones) in the same patient (defining the concept of FC), make it difficult to use biopsy and histopathological examination as a diagnostic and follow-up method. As a result, it is increasingly necessary to deepen and develop non-invasive diagnostic techniques in the diagnosis, staging, and monitoring of the effectiveness of the treatment of these lesions. Dermoscopy, High Frequency Ultrasonography (HFUS), Dermoscopy-Guided High-Frequency Ultrasound (DG-HFUS), Optical Coherence Tomography (OCT), and Reflectance Confocal Microscopy (RCM) are the most frequently used non-invasive techniques for the evaluation, treatment, and monitoring of precancerous and cancerous lesions, and each of them will be discus in the article.

The selection process for the literature included in this review was performed using different searching key-words such as actinic keratosis, field cancerization, non-invasive diagnosis, dermoscopy, reflectance confocal microscopy, ultrasonography, and treatment follow-up. The searching process and the selection of included articles was performed using PubMed, Web of Science, and Scopus databases by three dermatologists with experience in the analyzed techniques.

This review proposes, in addition to describing the main clinical and histological aspects of AK, an overview of the most frequently encountered skin imaging methods and their key diagnostic aspects currently used in the evaluation of AK and FC areas.

## 2. AK—Classification Systems and Clinical, Histopathological, and Therapeutic Aspects

### 2.1. Clinical and Histopathological Aspects

In 1999, Yantsos introduced a histological categorization for AKs that sorts the lesions by the extent of atypical cells within the epithelial layer. Low-grade histological lesions, termed KIN1, exhibit atypical cell distribution confined to the lower third of the epidermis and are generally considered to have a minimal risk of turning into cancer. In contrast, lesions categorized as KIN2 and KIN3 are of higher histological grades. They show atypical cells extending into the lower two-thirds and across the full thickness of the epidermis, respectively, and are associated with a higher likelihood of malignant transformation [15].

Clinically, AK lesions are graded according to a system by Olsen et al. which categorizes them from grade 1 (faint, flat pink spots that are easier to feel than see) to grade 3 (thick, scaly, hyperkeratotic lesions that may resemble early squamous cell carcinomas [SCCs]) [16]. This clinical classification is widely used but has its limitations, with recent studies indicating a discrepancy between Olsen’s clinical grades and histological severity, with the former often underestimating the latter in 62.4% of cases [17]. The progression from AK to squamous cell carcinoma (SCC) is a recognized pathway, with untreated AK potentially transforming into SCC [18]. Genetic analysis has revealed similar UVB-related mutations in both AK and SCC, notably in the p53 gene [19]. UVB impacts the basal layer of the epidermis, causing DNA damage that can lead to SCC, whereas UVA contributes to damage through the generation of free radicals [9,20]. Additionally, chronic sun exposure may induce a state of immune suppression within the epidermis [21].

This clinicopathological classification system for actinic keratosis (AK) progression stages proposed by Rowert-Huber in 2007 was designed to highlight that all AK lesions are essentially in situ SCC from the onset, but at different stages of progression, as can be seen in Table 1 [22]. When multiple grades appear within the same lesion, the highest grade determines the stage. The system also notes that it is not possible to predict which lesions will become invasive, leading to the conclusion that all AK lesions should be treated.

“IDRBEU” is another clinical system used for AK classification. It includes the following parameters, as can be seen in Table 2 [23].

Further research by Fernandez-Figueras and others has revealed that AK, even in early stages, has the potential to progress to invasive cutaneous SCC without necessarily passing through intermediate stages of keratinocyte intraepidermal neoplasia, challenging the traditional view of AK development [24,25]. This suggests that all AK lesions and the broader field cancerization area should be considered for treatment due to potential subclinical changes and genetic mutations present, similar to AK. It has also been highlighted that the hair follicle might play a significant role in the progression to deeply invasive cutaneous SCC, linked to the spread of abnormal keratinocytes in AK [26]. These insights are shaping clinical approaches to AK, with a European consensus advocating for the treatment of both individual AK lesions and field cancerization, tailored to the patient’s individual needs and characteristics [27].

### 2.2. Therapeutic Aspects

Treatments are chosen based on the AK’s clinical traits, the extent of the area needing treatment, the proven effectiveness, the tolerability of the treatment, and patient preference [28]. The goal is to prevent or remove lesions in accordance with the “field cancerization” concept, which posits that UV-damaged areas are prone to develop multiple AKs or lesions with a higher risk of progressing to SCC [29]. Areas showing poikiloderma or other signs of extensive sun damage should be addressed [30,31]. Treatment modalities include broad (“field-directed”) approaches for widespread damage and more concentrated (“lesion-specific”) treatments for individual lesions. The tractability of AK may vary with location. AKs on the upper limbs often prove more challenging to treat than those on the scalp or face [32,33]. A summary of the treatments for AKs can be found in Table 3 in the next chapter.

#### 2.2.1. Topical Treatments

Topical treatments for AK can include various topical agents such as imiquimod, 5-fluorouracil (5-FU), diclofenac, piroxicam, retinoids, tirbanibulin, and ingenolmebutate.

Imiquimod functions as an immune-response modifier by acting as an agonist to Toll-like receptor (TLR)-7. It is authorized in strengths of 5%, 3.75%, and 2.5% for treating AK on the face or scalp of patients with normal immune systems. Studies suggest that the 3.75% and 2.5% formulations offer considerable benefits over the 5% concentration by providing similar effectiveness while minimizing adverse reactions and reducing the length of treatment. These lower concentrations also permit treatment over a larger surface area, such as the entire face or a balding scalp [34,35]. Due to the high frequency of local skin reactions caused by its mechanism, imiquimod treatment is not advised during the summer months and is not the preferred option for patients with weakened immune systems [36]. Its effectiveness varies among studies, impacted by factors such as the concentration, frequency, and duration of applications; some authors report complete clearance in 48.3–52.3% and partial responses in 64–75.4% [37,38,39]. Furthermore, 5-FU is an antimetabolite family member, which works by inhibiting thymidylate synthase, essential for DNA synthesis and RNA functions. Available in various formulations, 5-FU can be applied in concentrations ranging from 0.5% to 5%, once or twice a day for 2 to 12 weeks. Its efficacy varies, with a full clearance rate of 96% seen with the 5% cream at 8 weeks, compared to a 48% clearance with the 0.5% cream [40,41].

Diclofenac, a nonsteroidal anti-inflammatory drug, acts by inhibiting cyclooxygenase 2, thus interfering with tumor growth, cell proliferation, angiogenesis, and apoptosis prevention [42]. It is available as a 3% gel combined with 2.5% hyaluronic acid, designed to treat AK on the face and scalp by addressing clustered lesions and FC. The recommended use is twice a day for 60–90 days, with a complete clearance rate of 41% at the end of treatment, increasing to 58% at a 30-day post-treatment assessment [42].

Piroxicam, another nonsteroidal anti-inflammatory drug similar to diclofenac, is a strong cyclooxygenase-1 inhibitor and also impedes proteins involved in tumor proliferation [43]. A topical formulation that combines 0.8% piroxicam with a broad-spectrum sunscreen, when used twice daily for six months, showed significant improvement in AK treatment outcomes. A complete response was seen in 55% of patients, with the results sustained at a 1-year follow-up [43].

Retinoids represent another treatment option for AKs and in situ cSCC [44]. Ianhez et al. reviewed studies on retinoids use for AK, covering both systemic and topical forms across different generations [44]. Clinical results vary, with some trials showing high efficacy and others showing low. However, evidence significantly supports the use of topical tretinoin and adapalene for AK, with 0.1% tretinoin and 0.1–0.3% adapalene being beneficial, albeit off-label [45,46]. Combination treatments like retinoids with 5-fluorouracil show more robust results compared to retinoid alone [47]. Photodynamic therapy coupled with adapalene pretreatment has proven highly effective for AKs on upper extremities [48].

Tirbanibulin, which acts as an inhibitor of Src kinase unrelated to adenosine triphosphate, prevents the polymerization of tubulin in cells undergoing division [49].

In 2020, the FDA approved tirbanibulin 1% ointment for treating non-hyperkeratotic, non-hypertrophic Olsen grade I actinic keratosis (AK) on the scalp and face, covering a 25 cm^2^ area, with a regimen of daily application for five consecutive days [49,50]. Tirbanibulin’s effectiveness stems from its ability to temporarily bind to and inhibit tubulin polymerization, leading to microtubule disruption. This action results in cell cycle arrest and apoptosis in proliferating cells. Moreover, tirbanibulin disrupts Src tyrosine kinase signaling, which is significantly involved in the development of both AKs and squamous cell carcinoma (SCC) [51].

Tirbanibulin distinguishes itself from other actinic keratosis (AK) treatments by not causing tissue necrosis, and it has a low incidence of severe local skin reactions (LSRs). This characteristic contributes to its high tolerability and safety profile. Campione et al. conducted a study demonstrating that a five-day regimen of daily tirbanibulin application resulted in a complete clinical and dermoscopic response in 70% of the participants by day 57 [52].

Ingenolmebutate stands out among topical medications for the field treatment of AKs due to its dual-action mechanism that boosts its effectiveness, leading to the destruction of lesions within 2–3 days [53,54]. Its therapeutic impact is a result of its role as an agonist for protein kinase C (PKC) within cells, which prompts swift necrosis of the affected cells and initiates neutrophil-driven antibody-dependent cellular cytotoxicity (ADCC) [55].

Resiquimod is a topical immune-response modifier that acts on Toll-like receptors 7 and 8 (TLR7 and TLR8) to activate myeloid and plasmacytoid dendritic cells. It is believed to be more potent (10–100 times) than imiquimod in treating AK due to its ability to induce higher levels of interleukin-12 and tumor necrosis factors. A phase II study explored four concentrations (0.01%, 0.03%, 0.06%, and 0.1%) of resiquimod gel, with complete clearance rates ranging from 77.1 to 90.3% [56].

#### 2.2.2. Physician-Managed Treatments

Physician-managed treatments include cryosurgery, chemical peels, photodynamic therapy (PDT), laser therapies, deep shave, or surgical excision.

Cryotherapy with liquid nitrogen is commonly used in an office environment for targeting specific lesions and can be administered without anesthesia; this method is usually well-received by patients [57]. Cryosurgery’s success rate in completely clearing lesions varies between 39 and 76%, and it has been observed that combining cryotherapy with other topical treatments enhances its efficacy [58]. Specifically, when used in conjunction with other topical agents (such as 5-FU cream, 5-FU-SA, imiquimod, or diclofenac in 2.5% hyaluronic acid) or associated with photodynamic therapy (PDT), the complete response rate improves to between 73.2 and 89%, while maintaining a similar tolerance level to cryotherapy alone [59,60,61,62].

PDT is a low-invasive technique that aims to induce apoptosis and necrosis of malignant cells selectively. By using photosensitizers like 5-Aminolevulinic acid (5-ALA) and its methyl ester 5-Methyl aminolevulinate (5-MAL), which preferentially accumulate in abnormal epidermal keratinocytes, the treatment induces the formation of reactive oxygen species that cause direct cellular damage, including necrosis or apoptosis, and harm to vascular endothelial cells [63,64]. Conventional PDT (c-PDT) has proven to be an effective choice, especially for treating FC, with better results noted on the face and scalp (69–93% clearance) compared to the forearm and hands (44–80%) [65]. Daylight PDT (dl-PDT), which involves applying a photosensitizer and then exposing the area to daylight for two hours, has been found to yield benefits similar to c-PDT [64,66,67]. Reports indicate that dl-PDT can achieve complete clearance rates of 70–89%, with a comparatively low 12-month recurrence rate of 8.7 to 13% [68,69].

Chemical peels are a strategy used to address multiple or closely grouped lesions and FC [70,71]. A recent meta-analysis reviewing eight studies evaluated the efficacy of different chemical peelings. It was found that the combination of trichloroacetic acid (TCA) with Jessner’s solution is less efficient compared to monotherapy with 5-FU 5% cream. Conversely, the 5% 5-FU-70% glycolic acid solution showed significantly better results in treating AK than monotherapy with 70% glycolic acid alone. Monotherapies with 50% and 30% TCA were found to be less effective than c-PDT. [72].

Laser treatments can be also useful in the management of individual AK lesions and FC by removing the layers of the skin (epidermis, superficial dermis) that contain the precancerous lesions, or by using the procedure called full-face skin resurfacing. The use of carbon dioxide lasers for the full ablative procedure is considered to have better results than fractional ablative techniques, despite longer healing times. [66]. While larger clinical trials are necessary, the current literature suggests that lasers used in monotherapy have a success rate that is lower than PDT and comparable to topical treatments with 5-FU and 30% TCA in efficacy [73].

Surgical excision or biopsy is advised for AK when there is diagnostic uncertainty or when lesions are resistant to treatment [74]. Histological examination can confirm or exclude the presence of invasive SCC, and deeper shave or surgical excision is recommended to assess for any dermal invasion [63].

**Table 3 cancers-16-00484-t003:** Summary of treatments for AKs.

Treatment Category	Treatment Type	Mechanism of Action	Application and Efficacy
Topical treatments	Imiquimod	Immune-response modifier, TLR-7 agonist	5%, 3.75%, 2.5% strengths for face/scalp. Lower concentrations reduce adverse reactions and allow broader application [34,35].
	5-Fluorouracil (5-FU)	Antimetabolite, inhibits thymidylate synthase	Applied 0.5% to 5%, 1–2 times/day for 2–12 weeks. Up to 96% clearance with 5% cream [40,41].
	Diclofenac	NSAID, inhibits cyclooxygenase 2	3% gel with 2.5% hyaluronic acid. Twice daily for 60–90 days. Up to 58% clearance post-treatment [42].
	Piroxicam	NSAID, cyclooxygenase-1 inhibitor	0.8% formulation with sunscreen, twice daily for six months. 55% complete response rate [43].
	Tirbanibulin	Inhibitor of Src kinase	FDA-approved for topical treatment of AK on face or scalp [49,50].
	Ingenol Mebutate	Dual-action, PKC agonist	Destroys lesions within 2–3 days [53,54].
	Resiquimod	Immune-response modifier, TLR7, and TLR8 agonist	Phase II study showed 77.1–90.3% clearance [56].
Physician-Managed Treatments	Cryosurgery	Liquid nitrogen application	Targets specific lesions, success rate 39–76%. Enhanced efficacy with topical treatments [58].
	Photodynamic Therapy (PDT)	5-ALA or 5-MAL as photosensitizers	c-PDT and dl-PDT. 69–93% clearance on face/scalp, 44–80% on forearms/hands [65,66].
	Chemical Peels	TCA with Jessner’s solution, or 5-FU with glycolic acid	Varied efficacy, less effective than c-PDT in some cases [72].
	Laser Treatments	Removes epidermis and papillary dermis	Fully ablative carbon dioxide laser more effective than fractional ablative [73].
	Surgical Excision	Biopsy for diagnostic confirmation	Recommended for uncertain diagnosis or treatment-resistant lesions [74].

AK—actinic keratosis, TLR-7—Toll-like Receptor 7, NSAID—nonsteroidal anti-inflammatory drugs, PKC—protein kinase C, TLR-8—Toll-like Receptor 8, 5-ALA—5-Aminolevulinic acid, 5-MAL—5-Methyl aminolevulinate, c-PDT—conventional photodynamic therapy, dl-PDT—day-light photodynamic therapy, TCA—trichloroacetic acid.

## 3. Non-Invasive Evaluation of AK

### 3.1. Dermoscopy of AK

Dermoscopy is a noninvasive diagnostic technique used for the evaluation of pigmented and nonpigmented skin lesions, for both diagnosing and evaluating the treatment response [75]. It enhances the detection of skin cancer and improves the frequencies of correct diagnoses and proper treatments. It can amplify the image in order to see the morphological structures in epidermis and papillary dermis which normally are not seen with the naked eye.

Using standardized algorithms based on pattern analysis, dermoscopy increases the diagnostic accuracy of melanocytic and non-melanocytic skin lesions [76]. Regarding the sensitivity and specificity of dermoscopy for the evaluation of skin tumors, studies have shown a more appropriate management and a correct diagnosis when using the dermoscope in comparison with naked eye examination. Also, more confidence in the diagnosis when using dermoscopy has shown an improvement in management choices [75].

Actinic keratoses have characteristic dermoscopic features (Table 4) which can be useful in the clinic for the diagnostic and follow-up during and after treatment management.

Overall, the dermoscopic pattern of AK is the “strawberry” appearance which is characterized by a red pseudo-network formed by erythema in the background, interrupted by the prominent follicular or adnexal openings (with a white halo that corresponds to the follicular openings and a yellow keratotic plug, forming a target lesion) (Figure 1) [77].

Pigmented AK lesions reveal a pigmented pseudo-network in between the follicular openings with a targetoid shape, sometimes associated with a red pseudo-network. Other dermoscopic signs evaluate surface scale (homogeneous white to yellow or brown areas), linear wavy vessels (surrounding the hair follicle, with wavy, curved shape), and structures that are visible only in polarized-light dermoscopy. Among those, the rosette sign (four closely aggregated white, small dots correlated with the follicular opening looking like a four-leafed clover—if we try to connect the four dots with a line, it can form a rhombus) and shiny streaks (perpendicular, white lines with a length of a few millimeters) are frequently encountered. Red “starburst pattern” with lines arranged in a radial pattern or hairpin vessels surrounding an area of yellow-to-white scales are criteria that suggest the transformation of AK in a malignant lesion [77,78].

Dermoscopy remains a noninvasive promising technique for the diagnosis of AK, possibly overcoming the lack of precision and limits of naked eye diagnosis that requires histopathological biopsies [78]. It is also a useful tool in monitoring the clinical response to treatment and the recurrence rate. For that, using dermoscopy allows the clinician to individualize the treatment in order to achieve clinical remission and to determine the end point of the treatment [80].

Even though the accuracy to recognize benign and malignant skin tumors by the clinician is increased when using dermoscopy, false positives and negatives do exist [81].

Limitations of this method may arise when dermoscopy is used by a non-experienced user, regarding the diagnostic or in connection with the management decision of the malignant lesions. The importance of experience and practice has been highlighted in studies for pigmented and nonpigmented skin lesions. Otherwise, the sensitivity and specificity of the method used by a non-experienced practitioner is equivalent to clinical examination alone [82,83].

Furthermore, newer noninvasive diagnostic techniques including reflectance confocal microscopy (RCM) have been shown to reach similar or even higher diagnostic values compared to dermoscopy [84].

### 3.2. RCM Diagnosis of AK

RCM is an emerging noninvasive imaging method that has been effective as a diagnostic tool for various skin conditions. It serves as an intermediate modality between dermoscopy and traditional histology, providing a side-by-side view of skin lesions and offering in vivo visuals of the epidermis and the top layers of the dermis with a resolution close to that of histopathological samples [85,86].

Clinically distinguishing between AKs, SCCs in situ, and their variants can be difficult. Additionally, skin biopsy, a common method for diagnosis, is invasive and complicates the monitoring of the same lesion post-treatment. However, these challenges might be mitigated through the use of RCM.

As expertise in the use of RCM for non-melanocytic skin cancers grows, the technique is increasingly recognized for its potential benefits. Despite the often clinically similar presentations of SCC and AK, accurate differentiation between these conditions remains challenging. Reliance solely on clinical observations for diagnosis is frequently unreliable. Biopsies, while diagnostic, are invasive and sometimes limited in their effectiveness, mainly due to potential sampling errors [87]. Moreover, RCM is also useful in assessing the lateral border of the lesions in order to improve the surgical outcome [88].

A key limitation of RCM in diagnosing skin lesions is its restricted depth penetration, particularly in hyperkeratotic lesions, often associated with invasive SCC [89]. This limitation hinders RCM’s ability to evaluate the dermo-epidermal junction and basal layer, both crucial layers for diagnosing potentially malignant hyperkeratotic conditions. Overcoming this challenge might involve removing the hyperkeratotic scale through curettage or using keratolytic agents before RCM imaging [90].

Consequently, distinguishing SCC from AK using RCM primarily depends on identifying extensive keratinocytes atypia through the whole thickness of the epidermis in SCC, as opposed to AK where disarrangement and atypia, if present, are usually confined to the stratum granulosum. In SCC, nest-like structures and pleomorphic cells may be observable in the dermis, features that are not present in AK [91].

Moreover, distinguishing between varying extents and severities of atypia can be challenging. This requires skilled RCM observers and high-quality image acquisition. A study by Pellacani et al. highlighted the correlations between RCM and histology in assessing the degree of atypia for AK lesions [91]. Nevertheless, while dyskeratotic cells were observed in both SCCs and AKs, there remains a lack of clear definitions for their morphological characteristics in RCM imaging. This underscores the need for standardized definitions of RCM features to enhance the consistency and accuracy of image assessment [92].

The primary characteristics of AK observed through RCM include surface irregularities, architectural disorder, and cellular variability within the spinous and granular layers. These features provide a sensitivity of 80% and a specificity of 98.6% in identifying AK [89]. RCM offers a horizontal view of the skin layers, which means that it cannot capture vertical sections of the lesion [89].

AK typically manifests with surface scaling, parakeratosis, and an uneven honeycomb pattern in the spinous-granular layers. At the dermo-epidermal junction (DEJ), small, regular dermal papillae are often seen [93,94]. More RCM characteristics of AK are presented in Table 5 and Figure 2 and Figure 3.

Zalaudek et al. provided initial findings linking RCM and dermoscopic grading AK (Table 6) [96]. In Grade 1 AK, there are localized regions of atypical honeycomb patterns at the stratum spinosum level, mixed with areas showing the normal, typical honeycomb pattern. Grade 2 AK exhibits more widespread atypical features across both the stratum spinosum and granulosum, with keratinocytes displaying pronounced irregularities in size and shape. Grade 3 AK is distinguished by a significantly abnormal honeycomb pattern, along with sections where the normal epidermal layers are partially disrupted, referred to as a disarranged pattern. This stage also features highly variable and irregularly shaped keratinocytes, as well as inconsistent connections between them. Table 7 encapsulates the grading of AKs based on their appearance under RCM and dermoscopy, as described by Zalaudek et al. [96].

Various studies have shown a decrease in atypia of epidermal layers in AKs following treatment with topical imiquimod and PDT. Ulrich et al. observed lingering atypia in two AK lesions (one without clinical response and another with apparent clinical disappearance of the lesion) following imiquimod topical application, indicating possible incomplete clearance of the lesion. The authors also created an RCM atypia score to assess cellular changes in those lesions post-treatment, thus enhancing RCM’s utility [97]. They concluded that using RCM to detect early residual atypia could potentially reduce recurrence rates [97].

In two case-control studies, the AK lesions were compared with apparently clinically normal skin located in the proximity of those lesions or normal skin from the contralateral side. However, the widespread neoplastic changes in sun-exposed skin caused by UV light may cause visible changes in RCM, even in apparently clinical normal skin, making the areas chosen for controls not the best choices. Jafari et al. found discrete atypia at the stratum spinosum level, dilated blood vessels in normal skin located in the proximity of AK and solar elastosis [95]. Therefore, a more reliable control group might be the patients without a history of UV exposure and UV-induced neoplasia [98].

The study released by Rishpon A. et al. revealed the criteria for diagnosis SCCs and AKs, and the correlation between dermoscopy and in vivo RCM [89]. Scales are a common feature of benign skin lesions such as psoriasis plaques, thus being a non-specific characteristic for AKs and SCCs, limiting its diagnostic value when viewed in isolation. Scales correspond to orthokeratosis or parakeratosis in histology. In this study, scales were identified in 92% of the dermoscopic images and in 95% of the RCM mosaic images. Polygonal nucleated cells in the stratum corneum, indicative of parakeratotic cells, were found in only a few lesions in this study. This limited observation may be due to the constraints of our visual analysis, influenced by the resolution limits of the RCM technology [89].

All cases of AKs and SCCs exhibited either an atypical honeycomb pattern or a disrupted epidermal arrangement. These patterns are most clearly identifiable under high magnification, specifically in RCM individual images. There is no known correlation between these features and any clinical or dermoscopic features, and these features are distinct to RCM’s evaluation of SCC. The SCCs demonstrated pronounced atypical features and/or disorganization in the spinous-granular layer, while most AK samples showed either a localized disruption or a slightly abnormal honeycomb pattern [89].

Round, bright, nucleated cells in the spinous-granular layer represent an RCM indicative of atypical and dyskeratotic keratinocytes found in both SCCs and AKs during histopathologic analysis, being more common in SCCs than in AKs. Rishpon A. et al. highlighted that the frequency of this characteristic was in 55% of the total cases [89].

Dermoscopy tends to have less detection of vessels in comparison with RCM, due to RCM’s infrared laser light penetrating deeper than the visible light of the dermoscope. Moreover, vessel visibility might be impacted by pressure during contact dermoscopy. Dermoscopic criteria of dotted and glomerular vessels was correlated with round blood vessels in the superficial dermis on RCM. AKs display fewer round vessels in the superficial dermis on RCM, compared with SCCs [99].

Research has explored using RCM to monitor treatment responses in AKs. In 2018, Ishioka et al. observed a decrease in polygonal nucleated cells and isolated keratinocytes using 5% FU cream, as measured by RCM [100]. Pasquali et al. evaluated the evolution of AK treatment with a combination of cryotherapy–ingenol mebutate, identifying the following indicators of treatment response in RCM: parakeratosis, honeycomb pattern, round papillary vessels, and detachment of the stratum corneum were reliable indicators of RCM [101]. In 2019, Benati et al. reported notable improvements in keratinocyte disarray, parakeratosis, and crust formation after using topical imiquimod, as seen through RCM [102]. These studies collectively affirm the utility of RCM in tracking AK treatment outcomes. However, they also highlight significant variations in methods for assessing these responses. Similar inconsistencies in identifying reliable RCM parameters are evident in baseline, non-therapeutic AK characterization studies [103,104]. Establishing a standardized, widely accepted protocol for evaluating AK treatment responses via RCM could harmonize results across different studies and enhance the clinical value of ongoing patient monitoring.

Current studies show promising results for RCM diagnostic accuracy, with sensitivities and sensibilities ranging from 79–100% and 78–100%, respectively, in diagnosing AK, SCC in situ, and SCC. A randomized control study that compares RCM with histological examination in suspected SCC cases could determine RCM’s diagnostic accuracy and is of interest [105].

**Table 7 cancers-16-00484-t007:** Grading of AK from a clinical, dermoscopic, in vivo RCM, and histopathological perspective.

AK Grade	Clinical Appearance [22]	Dermoscopic Appearance [77]	In Vivo RCM Appearance [96]	Histopathological Appearance [106]
1	Barely palpable AK (more evident on palpation than visual examination)	Pseudo-network red pattern	Focal areas with atypical honeycomb pattern at the stratum spinosum level	Focal keratinocyte atypia in the lower third of the epidermis
2	Moderately thick AK (easily observable visually and on palpation)	Keratotic follicular openings on an erythematous background	Diffuse keratinocyte atypia involving the spinous and granular layers; marked keratinocyte atypia with varying sizes and shapes	Focal keratinocyte atypia in at least the lower third of the epidermis; focal hyperkeratosis alternating with orthokeratosis and parakeratosis; prominent acanthosis and keratinocyte buds in the upper papillary dermis
3	Thick AK with hyperkeratotic appearance	White-yellowish astructural zones	Disorganized honeycomb pattern characterized by the presence of pleomorphic keratinocytes and irregular interkeratinocyte connections	Diffuse atypical keratinocyte proliferation involving the entire thickness of the epidermis; parakeratosis, acanthosis, papillomatosis, involvement of annex structures

AK—actinic keratosis, RCM—reflectance confocal microscopy.

### 3.3. Optical Coherence Tomography (OCT) and AK

Optical Coherence Tomography (OCT) offers non-invasive, real-time imaging of the skin, operating similarly to ultrasound, but using near-infrared light impulses instead of sound waves. OCT functions on the principle of interferometry, capturing images by measuring the intensity of light reflected at varying depths. The images produced by OCT can be cross-sectional (vertical) or en-face (horizontal), and they may be two- or three- dimensional. Standard OCT systems typically achieve a penetration depth of 1–2 mm and a resolution of less than 7.5 μm laterally and less than 5 μm axially, covering a 6 mm × 6 mm field of view. Although offering high resolution, conventional OCT is limited to imaging larger structures and patterns rather than individual cells. It is particularly effective for identifying AK, as it can penetrate to the dermo-epidermal junction (DEJ), revealing sweat glands, hair follicles, and other underlying structures up to the mid-dermis [107].

High definition OCT (HD-OCT) is a variant of standard OCT, adhering to the same basic principles but with enhanced capabilities. It boasts a lateral and axial resolution of 3 um, enabling it to visualize individual skin cells. The field of view for HD-OCT is smaller, at 1.8 × 1.5 mm, and its images are presented in both cross-sectional (B-scan) and en-face modes. The en-face mode is more extensively analyzed in HD-OCT than in conventional OCT. HD-OCT’s penetration depth is 570 um, situating it between RCM and conventional OCT in terms of penetration depth and image resolution [108].

Several studies have utilized OCT to image AKs, describing various morphological characteristics. These features include the disturbance of skin layers, the presence of white streaks and dots, hyperkeratosis, and an increased epidermal thickness [109,110,111].

The term “disruption of layers” or “architectural disarray” is used to describe the alteration or complete loss of the skin’s normal layered structure, often evidenced by a partial or total absence of the DEJ. White or hyper-reflective structures (streaks, dots) are commonly seen in AK lesions with scales (hyperkeratosis) [109]. However, these features may also be present in lesions characterized by crusted ulcers and scales. The presence of a dark band in the stratum corneum (SC) can be observed in some AK lesions [110].

In HD-OCT, AKs are characterized by a disordered epidermal structure and an atypical honeycomb pattern. This pattern is typically observed as a diminishing arrangement of normal polygonal cells throughout the epidermis when viewed in en-face mode. The term “atypical honeycomb pattern” is used to denote the presence of cells within the epidermis that vary in size, shape, and reflectivity. Many studies have noted a well-defined dermo-epidermal junction (DEJ), which appears in cross-sectional images as a small, dark, homogeneous band distinguishing the epidermis from the more reflective dermis [112].

Maier et al. conducted a study comparing the morphological HD-OCT features and histopathological aspects of AK [113]. They discovered that HD-OCT effectively visualized histopathological features such as hyperkeratosis and parakeratosis, identifiable in HD-OCT as irregular disruptions in the SC and the thickening of the SC. These disruptions were described as SC disruption. However, histopathological inflammatory infiltrates in the upper dermis were not entirely discernible in the HD-OCT images. A key observation from this study was that HD-OCT en-face images could reveal cellular alterations of keratinocytes as heterogenicity in the morphology and distribution of acantholysis and dyskeratosis [113].

### 3.4. Ultrasound and High Frequency Ultrasound Imaging in AK

Ultrasonic imaging has been part of dermatological practices for almost three decades [114]. Its introduction dates back to 1979 when Alexander and Miller first used ultrasonography as a non-invasive method to determine the thickness of normal skin. During the 1980s and 1990s, high-resolution ultrasonography became a tool for non-invasive evaluation of skin nodules and various skin diseases [115,116]. Traditionally, the diagnosis of most skin conditions, whether localized or widespread, has predominantly depended on physical examination [117]. However, studies indicate that HFUS offers significant advantages over clinical examination alone, providing essential insights for detecting and precisely measuring numerous clinical and subclinical skin lesions [118].

Dermatologists have specific requirements for non-invasive ultrasonic skin examinations, including determining the size, shape, structure, and depth of penetration of skin lesions. USG offers a non-invasive approach that allows for the live, real-time histological evaluation of skin structure and specific conditions [119]. In assessing skin lesions, HRUS, combined with color Doppler, proves to be a safe, economical, and repeatable diagnostic method. Its ability to provide detailed images makes it a valuable alternative to more invasive procedures like fine needle aspirations and biopsies [120].

The Dermoscopy-Guided High-Frequency Ultrasound (DG-HFUS) technique integrates dermoscopy with HFUS for concurrent imaging of both the skin’s surface and its full depth. This method utilizes a single-element ultrasound transducer, operating at a nominal center frequency of 33 MHz. Two-dimensional ultrasound imaging is achieved through mechanical scanning, which involves moving a physical slider on the device, coupled with an automated scan conversion algorithm [121].

Optical guidance is employed by the device to enhance the accuracy and consistency of the recordings. In practical terms, this translates to the real-time display of optical images of the scanning area, which are magnified 10 times, during the positioning phase before scanning begins. The outcomes of a single recording session are presented as a two-dimensional ultrasound image alongside the corresponding optical dermoscopic image of the skin’s surface (Figure 4 and Figure 5). This includes a marker to show the ultrasound image slice’s relative position on the surface. To improve contrast and aid in the identification of skin structures, the ultrasound image uses a color scale instead of the traditional grayscale. This feature enhances the visibility and differentiation of various skin elements within the image [122].

Non-melanoma skin cancers (NMSCs) usually appear on ultrasounds as hypoechoic, heterogeneous tumors with irregular borders. Compared to Basal Cell Carcinoma (BCC), SCC is more aggressive and more likely to infiltrate deeper tissues and regional lymph nodes, necessitating thorough ultrasound examination. SCC typically exhibits more vascularity than BCC, showing an increased number of blood vessels within the lesion [123]. Doppler ultrasound can reveal low-resistance, pulsating blood flow either within the tumor or around its edges [117]. However, in certain instances of squamous cell carcinoma, the presence of a hyperkeratotic epidermis might completely reflect the ultrasound waves, rendering it impossible to accurately measure the tumor’s thickness [124].

In recent times, non-invasive methods like dermoscopy, reflectance confocal microscopy (RCM), and optical coherence tomography (OCT) have been developed to effectively monitor and diagnose actinic keratoses (AKs). However, their high costs and limited depth of penetration, which may result in incomplete lesion assessment, restrict their widespread use in everyday clinical practice. In contrast, high-frequency ultrasound (HFUS) is advantageous due to its affordability, ease of use, repeatability, real-time imaging capability, non-invasiveness, and versatility. HFUS can provide detailed images of skin layers and deeper structures, making it a valuable tool for diagnosing suspected skin cancers and aiding in treatment decisions.

## 4. Field Cancerization and the Role of Non-Invasive Imaging Techniques in Diagnosis and Treatment Follow-Up

The term of field cancerization (FC) was first proposed in the second half of the 20th century (1953) by Slaughter et al. and defined a surface of the epithelium (oral mucosa) that was subjected to carcinogenic factors which generated irreversible changes at the cellular level, initiating the transformation of some cells towards malignancy [117]. Thus, this notion appeared in order to explain the appearance of numerous multifocal neoplasia at the level of the oral mucosa, which were than considered secondary cancers rather than local recurrences [125].

Later on, FC was defined by the presence of molecular anomalies in an apparently normal tissue from a histological point of view, a concept that changed the way we look at the management of these patients, excisions with apparently “tumor free” margins no longer being curative in an FC area [126].

Although the concept of FC has been described at the level of several organs that could be exposed to external carcinogenic factors, the skin remains in permanent interaction with the external environment and implicitly with ultraviolet radiation (UVR), making it possible to study and easily observe the phenomenon of FC and its evolution [127].

Currently, multiple definitions have been used to describe this phenomenon at the skin level, the most recent one being Willenbrink’s definition who describes FC from a clinical perspective: “multifocal clinical atypia characterized by Aks or squamous cell carcinomas in situ with or without invasive disease, occurring in a field exposed to chronic UVR” [128].

From a molecular, genetic, and immunological point of view, FC changes at the skin level were represented by changes in intracellular signaling pathways, DNA alterations induced by UVR, and oxidative stress; changes in the ratio of signaling molecules involved in the innate immune response, as well as the downregulation of some essential tumor suppressor genes. Moreover, these changes were present not only in the AK and SCC skin samples, but also in the apparently normal perilesional skin, which was subjected to the photodamage process, supporting once again the continuum of normal–AK-SCC skin [31]. The importance of establishing a clear clinical definition of this pathological entity is essential for the practicing dermatologist because of its impact on the adaptation of treatment [129]. Thus, according to a group of experts of the Progressing Evidence in AK (PEAK) Working Group, a clinical definition was formulated in 2018, namely: “FC is clinically defined as the anatomical area with or adjacent to AK and visibly sun damaged skin identified by at least two of the following signs: telangiectasia, atrophy, pigmentation disorders and sand paper” [31].

The simple identification of an area of FC significantly changes the therapeutic approach compared to the therapeutic lines described above for isolated AK by associating field therapy techniques with localized destructive methods (in the case of presence of AK lesions). Moreover, the diagnosis of FC requires an active education of the patient in order to self-examine, even in the absence of AK-type lesions, due to the increased risk of the emergence of non-melanocytic skin cancers [31,129].

The identification of individual isolated AKs, as well as their transformation into SCC, is currently performed by dermoscopic examination of the lesion (by applying the criteria mentioned above). Moreover, the identification of clinical and dermoscopic criteria of photodamaged skin in the perilesional area (telangiectasias, atrophy, pigmentary changes, etc.) brings to the eye of clinician the diagnosis of FC [130]. However, in the absence of skin imaging methods, differentiating a chronic photodamaged skin and an FC area without the presence of AK lesions is almost impossible [131]. The use of non-invasive imaging methods allows not only the easy identification of FC, sometimes even in the precancerous stages or even before the appearance of AK, but also opens the possibility of an efficient and non-invasive therapeutic follow-up method [130].

RCM is an efficient, non-invasive, and rapidly performing method of assessing the degree of AK and FC, allowing the dermatologist to evaluate the effectiveness of topical therapies in the absence of tissue sampling. Moreover, in the case of FC, where multiple subclinical lesions are present, RCM allows the identification of keratinocytic disarray before becoming visible dermoscopically. This allows for the initiation of treatment at an early stage and increases its effectiveness [100].

In a study published by Ulrich et al. in 2015, they highlighted the efficacy of RCM in the diagnosis and follow-up of AK treated with a topical solution of 0.5% 5-fluorouracil and 10% salicylic acid on a group of eight patients with AK and FC [132]. RCM allowed more accurate evaluations of AK grade, observation of subclinical cellular disarray, and confirmation of its disappearance at a cellular level after treatment. This study once again emphasized the benefit and rapidity of this non-invasive method, especially since AK lesions are located in photo-exposed areas (especially the face, head, and neck), where it is difficult to perform repeated biopsies [132].

Another study published by Agozzino et al. evaluated the usefulness of RCM and dermoscopy on a group of 54 patients in the evaluation of topical treatments with 0.8 piroxicam and sunscreen with sun protection factor 50, applied twice a day for 6 months. In this study, the team calculated objective scores of RCM and dermoscopy [133]. Regarding the RCM assessment, it took into account the following aspects: presence of scales, alteration of keratinocytes, parakeratosis, changes in the shape of keratinocytes (polygonal aspect), presence of atypical honeycomb pattern, presence of inflammatory cells in the epidermis, increased vascularity, changes in the appearance, and distribution of collagen fibers (disarray), as well as the presence of inflammatory cells in the superficial dermis, with a maximum possible score of 11. The calculation of these scores before and after applying the topical treatment allowed not only the objective quantification of the therapy’s effectiveness, but also the direct visualization of its impact on each of the layers of the epidermis, as well as at the level of the dermis [133].

Another study conducted by Ishioka et al., which evaluated the accuracy of RCM in determining the effectiveness of topical therapy with 5-fluorouracil compared to the histopathological examination on 40 AK lesions, demonstrated a similar accuracy of the RCM examination performed by an experienced investigator when compared with the histopathological examination (*p* < 0.001) [100]. Thus, the study highlighted an accuracy of 83.8%, a sensitivity of 84.6%, and a specificity of 83.3% in the correct identification of AK with the RCM examination [100].

Although the RCM has a sensitivity and specificity close to the histological ones, it is difficult to evaluate the same parameters with regard to the dermoscopic examination, with most of the published studies until now not being able to establish an accuracy of the dermoscopic diagnosis for actinic keratoses, but rather describing the dermoscopic aspects most frequently encountered in these lesions (see Dermoscopy and AK subsection). The only study that evaluated the histological–dermoscopic concordance was the one conducted by Huerta Brogeras et al. in 2012, which evaluated 178 patients with AK [78]. The study found a diagnostic sensitivity of 98.7% and a specificity of 95% for dermoscopic exam [78].

Later, Xiaoqin et al. used dermoscopy in the monitoring of PDT treatment with 5-aminolevulinic acid on a group of patients with KA, highlighting the fact that dermoscopic follow-up with the aim of establishing the end of the treatment decreases the relapse rate (2.85% in the case of dermoscopically followed patients vs. 12.5% in the case of the control group) [80].

## 5. Conclusions

Actinic keratoses are responsible for a large impact on healthcare systems world-wide, both through the unpredictable potential of malignant transformation and the need for treatments with potential local complications. Non-invasive imaging techniques (dermoscopy, reflectance confocal microscopy) not only allow the identification of clear criteria to establish a diagnosis, but also offer the dermatologist the opportunity to closely follow-up the cancerization field, as well as the different stages of local treatment, with the aim of establishing an end-point of treatment. Although still at the beginning, ultrasonographic imaging techniques (high frequency ultrasonography and dermoscopy guided ultrasonography) look promising and are able to complement the formerly mentioned through a different visualization plane, as well as through better in-depth penetration. However, they are still new techniques at the beginning of their journey, for which studies on large groups of patients are needed in order to establish clear criteria for the diagnosis and staging of AK. At the present time, reflectance confocal microscopy remains the imaging technique with the closest specificity and sensitivity to histopathological examination in the diagnosis and monitoring of these lesions.

## Figures and Tables

**Figure 1 cancers-16-00484-f001:**
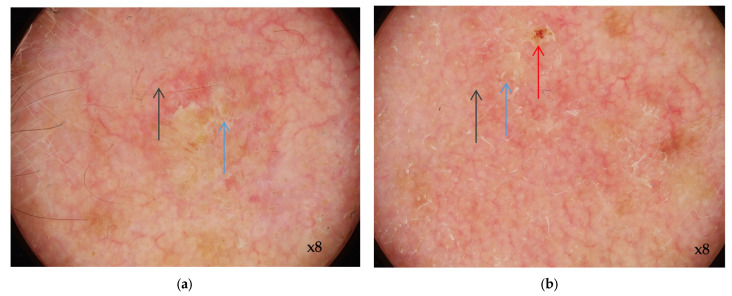
Dermoscopic aspects of actinic keratosis. Red pseudo-network with erythema in the background, interrupted by the follicular openings; yellow scales in the surface (blue arrows), slightly curved, wavy vessels (grey arrows) (**a**,**b**); erosion (red arrow) (**b**); Images from the personal library of VMV obtained with Casio DZ-D100 device (Dermoscopy Mode).

**Figure 2 cancers-16-00484-f002:**
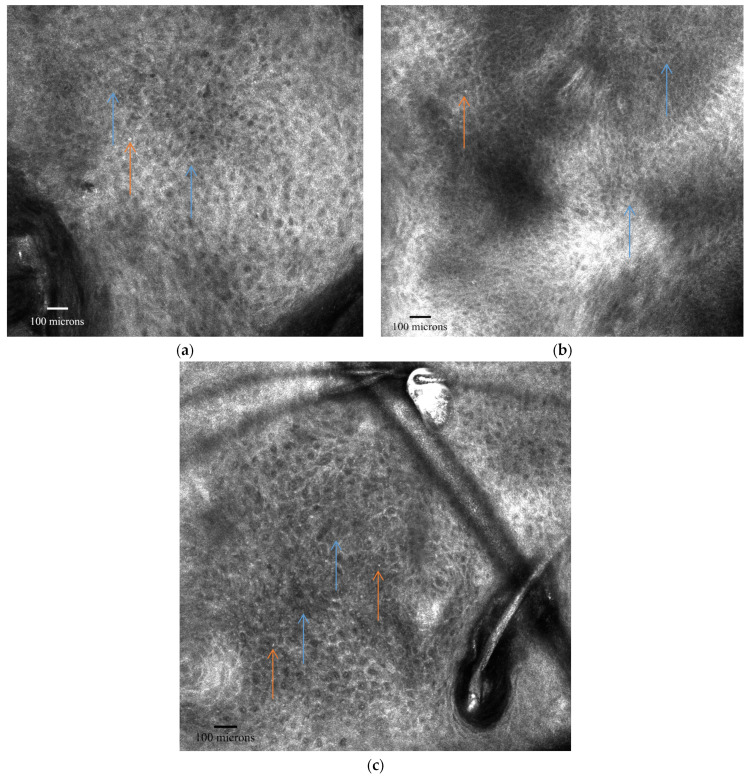
Reflectance confocal microscopy aspects of actinic keratosis at the level of spinous and granular layers—atypical honeycomb patterns with keratinocytes variable in size and morphology (blue arrows) (**a**–**c**); small hyperreflective cells corresponding to inflammatory cells (orange arrows) (**a**,**c**); Images from the personal library of VMV obtained with the VivaScope handheld 3000 device.

**Figure 3 cancers-16-00484-f003:**
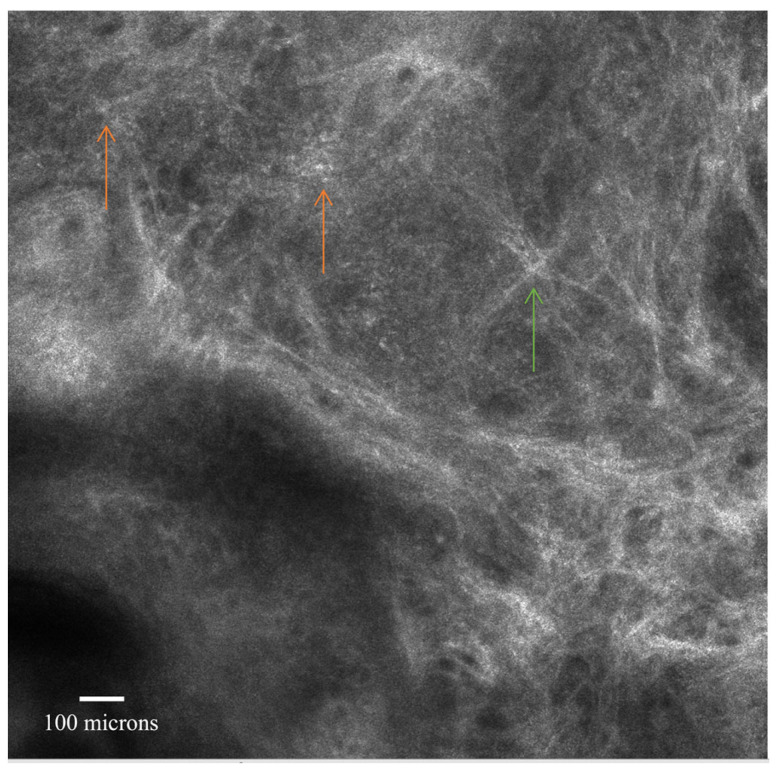
Reflectance confocal microscopy aspects of actinic keratosis at the level of dermis; hyperreflective collagen bundles which interconnects in a disarray pattern (green arrow); moderately refractive lace-like material adjacent to collagen bundles; small bright cells corresponding to inflammatory cells (orange arrows); dilated blood vessels. Images from the personal library of VMV obtained using a VivaScope handheld 3000 device.

**Figure 4 cancers-16-00484-f004:**
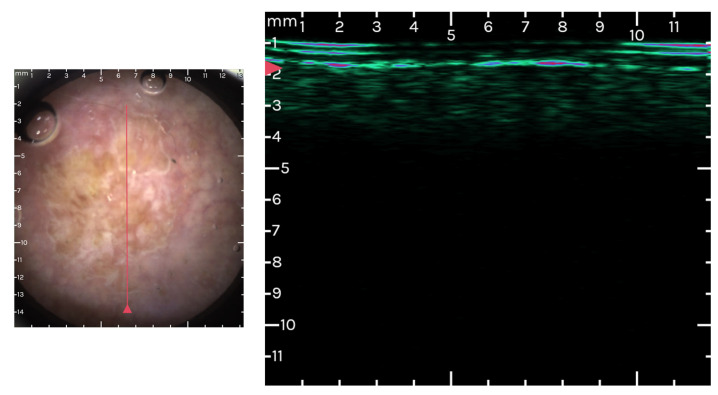
Dermoscopy-Guided High-Frequency Ultrasound images at the level of an actinic keratosis grade II–III. Images from the personal library of VMV obtained using the Dermus Vision Device (Skin Aid Software, 2023).

**Figure 5 cancers-16-00484-f005:**
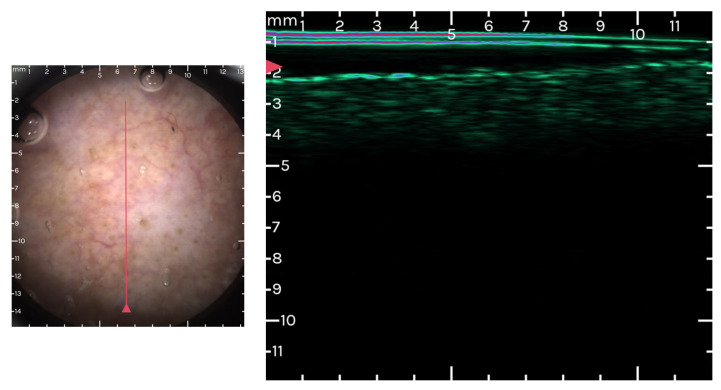
Dermoscopy-Guided High-Frequency Ultrasound images at the level of an actinic keratosis grade I–II. Images from the personal library of Voiculescu VM obtained using the Dermus Vision Device (Skin Aid Software, 2023).

**Table 1 cancers-16-00484-t001:** The clinicopathological classification system for AK progression stages [20].

Progression Stage	Description	Clinical Correlation
AK I	Initial phase of in situ SCC	Grade 1 — slightly palpable AK lesion (better felt than seen)
AK II	Intermediate phase of in situ SCC	Grade 2 — moderately thick AK lesion (easily felt and seen)
AK III	Advanced phase of in situ SCC	Grade 3 — Very thick, hyperkeratotic, and/or obvious AK lesion

AK I—actinic keratosis grade I, AK II—actinic keratosis grade II, AK III—actinic keratosis grade III, SCC—squamous cell carcinoma.

**Table 2 cancers-16-00484-t002:** “IDRBEU” clinical system [23].

Parameter	Description
I (induration/inflammation)	Presence of hardening or inflammation
D (diameter > 1 cm)	Lesion diameter larger than 1 cm
R (rapid enlargement)	Quick increase in lesion size
B (bleeding)	Occurrence of bleeding
E (erythema)	Redness of the skin
U (ulceration)	Formation of ulcers

**Table 4 cancers-16-00484-t004:** Diagnostic dermoscopic criteria for AK [76,77,78,79].

Dermoscopic Criteria	Description	Diagnostic
Erythema	areas of redness without a specific structure, lacking any areas of lighter pigmentation or distinct shape	AK I
Red pseudo-network	red regions without a defined structure, interlaced with small, round white spots that create a network-like pattern; these small white spots represent the openings of hair follicles	AK I
Strawberry pattern	red areas lacking a distinct structure, intermingled with round, target-like formations featuring an inner yellow and outer white ring that resemble strawberries; these target-like areas are associated with hair follicle openings filled with keratin	AK II
Rosettes	four white dots positioned in a diamond shape, creating a pattern like a four-leaf clover	AK I-II
Surface Scale	opaque areas ranging in color from yellow to light brown, lacking a defined structure and exhibiting a scaly or keratotic appearance, which occupy only a small portion of the tumor’s surface	AK III (in situ SCC)
Targetoid hair follicles with whitish halo	circular formations of varying sizes, featuring a center that is yellow to light brown and structureless, surrounded by a white, structureless outer ring; this pattern is indicative of keratotic plugs within the hair follicle openings in the skin	AK II SCC
Red starburst	red lines or hairpin-shaped vessels, lacking distinct structure and arranged radially, surrounding a yellowish white, structureless, scaly center, creating a pattern reminiscent of a starburst	AK II AK III (in situ SCC)
Erosions	irregularly spaced small areas ranging in color from orange to red and red-brown, without a defined structure; these areas are indicative of superficial bleeding and are typically found in conjunction with yellow, opaque structures	AK III (in situ SCC) SCC
Linear, wavy vessels	red structures that are linear or slightly curved, irregular in shape, size, and distribution	NPAK
Dotted vessels Glomerular vessels	closely packed, small red dots with a complex morphology, larger than typical dotted vessels and frequently grouped in clusters, representing a variation of the dotted vessel theme	AK III (in situ SCC)
Hairpin vessels	twisting and bending vascular loops, often encircled by a whitish halo, typically observed in keratinizing tumors	SCC
Rainbow pattern	lines with multiple colors (from blue to red)	AK SCC BCC

AK I—actinic keratosis grade I, AK II—actinic keratosis grade II, AK III-actinic keratosis grade III, SCC—squamous cell carcinoma, NPAK—non pigmented actinic keratosis.

**Table 5 cancers-16-00484-t005:** Key features of AK on RCM [95].

Skin Layer	RCM Features of AK
Stratum Corneum	Superficial disruption with large single keratinocytes Nucleated cells (parakeratosis) Small bright cells with dark centers (neutrophils)
Stratum Granulosum/Spinosum	Atypical honeycomb pattern: variations in cell size and morphology Broadened honeycomb pattern: areas with broadened and blurred intercellular connections Loss of regular epidermal architecture
Dermis	Slightly dilated blood vessels Solar elastosis: moderately refractive lace-like material adjacent to collagen bundles

AK—actinic keratosis, RCM—reflectance confocal microscopy.

**Table 6 cancers-16-00484-t006:** Grading of AKs as described by Zalaudek et al. [96].

AK Grade	Description
Grade 1	Focal areas of atypical, honeycombed pattern at the stratum spinosum level, mixed with areas of typical honeycombed pattern.
Grade 2	Diffuse atypia of keratinocytes in both the stratum spinosum and granulosum, with marked variation in cell sizes and shapes.
Grade 3	Markedly atypical, honeycombed pattern with partial disruption of normal epidermal layers (disarranged pattern), wide variability in keratinocyte size and shape, and irregular intercellular connections.

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
