# Peer review of "Digitally Enhanced Methods for the Diagnosis and Monitoring of Treatment Responses in Actinic Keratoses: A New Avenue in Personalized Skin Care"

_cancers, 2024, doi:10.3390/cancers16030484_

Round 1
Reviewer 1 Report
Comments and Suggestions for Authors
Dear Authors,
our review is very interesting and usefully for AK diagnosis.
So, I think what you will reduced treatment section and focused on new treatment . (so for tirbanibulin you can see //doi.org/10.3390/ph16121686.
For advanced tecnhiques you can also correlate with histological features.
Author Response
Dear Authors,
our review is very interesting and usefully for AK diagnosis.
So, I think what you will reduced treatment section and focused on new treatment (so for tirbanibulin you can see //doi.org/10.3390/ph16121686.
For advanced tecnhiques you can also correlate with histological features.
We acknowledge #Reviewer 1 for the comments that definitely helped us to improve the manuscript.
Thank you for this suggestion, we cited and discussed the suggested paper.
Reviewer 2 Report
Comments and Suggestions for Authors
Dear Author,
I read your manuscript concerning skin imaging methods to evaluate and monitor actinic keratosis. The paper is clear and easy to read and reports some information about non-invasive techniques, but some points must be addressed to improve the manuscript.
1. Text should be formatted according to the journal and APA styles.
2. Check references 113-114.
3. Grammatical and syntactic errors in the main files, correct.
4. Standardize the definition of actinic keratosis. In the abstract, it is reported that it is a precancerous lesion (an opinion I share), but in the introduction, AK is referred to as a squamous cell tumour.
5. Lines 63-65 are be unclear and redundant. Moreover, FC should be established in the introduction section and not just in the aim of the study.
6. In the introduction, non-invasive techniques should be reported in a few lines with their rationale for monitoring AKs.
7. References after Author et al. [], not at the end of the sentence.
8. In the treatment sections, I suggest reporting an interesting new study about tirbanibulin:
- Campione, E.; Rivieccio, A.; Gaeta Shumak, R.; Costanza, G.; Cosio, T.; Lambiase, S.; Garofalo, V.; Artosi, F.; Lozzi, F.; Freni, C.; et al. Preliminary Evidence of Efficacy, Safety, and Treatment Satisfaction with Tirbanibulin 1% Ointment: A Clinical Perspective on Actinic Keratoses. Pharmaceuticals 2023, 16, 1686. https://doi.org/10.3390/ph16121686
9. In the introduction section, specify that AK is a complex and multifactorial disease. I suggest reading and citing:
- Prezioso, C., Brazzini, G., Passerini, S., Di Fabio, C., Cosio, T., Bernardini, S., Campione, E., Moens, U., Pietropaolo, V., & Ciotti, M. (2022). Prevalence of MCPyV, HPyV6, HPyV7 and TSPyV in Actinic Keratosis Biopsy Specimens. Viruses, 14(2), 427. https://doi.org/10.3390/v14020427
10. The study design and methodology used for the research are unclear. Why did you not include Near-infrared spectroscopy? The material and method section is missing.
11. Retinoids are not reported in the article. Explain.
12. Table 5 format font.
13. In vitro, in vivo italicized, please.
Comments on the Quality of English LanguageMinor editing of English language required
Author Response
Dear Author,
I read your manuscript concerning skin imaging methods to evaluate and monitor actinic keratosis. The paper is clear and easy to read and reports some information about non-invasive techniques, but some points must be addressed to improve the manuscript.
- Text should be formatted according to the journal and APA styles.
Thank you for your observation. The text was filled in the Cancers template according with editor’s guidelines. We have modified the references in order to correspond with the guidelines indicated in the template.
- Check references 113-114.
Thank you for your observation. We have modified the reference number 114 and updated the numbers of the following references: 115-124.
- Grammatical and syntactic errors in the main files, correct.
Thank you for your observation. We have made several corrections.
- Standardize the definition of actinic keratosis. In the abstract, it is reported that it is a precancerous lesion (an opinion I share), but in the introduction, AK is referred to as a squamous cell tumour.
Thank you for your comment. Indeed, a typing mistake had occurred. We have corrected the definition in the introductory part:
“Actinic keratosis (AK), also known as solar keratosis, is a keratinocyte-derived precancerous lesion frequently found in adults, that typically develops on parts of the skin exposed to the sun.” (lines 38-39).
- Lines 63-65 are be unclear and redundant. Moreover, FC should be established in the introduction section and not just in the aim of the study.
Thank you for your observation. We have made the changes accordingly (lines 58-66).
- In the introduction, non-invasive techniques should be reported in a few lines with their rationale for monitoring AKs.
Thank you for your comment. We have introduced a short phrase with the main non invasive imaging techniques.
- References after Author et al. [], not at the end of the sentence.
Thank you for your observation. We have followed the guidelines from the Cancers template: In the text, reference numbers should be placed in square brackets [ ] and placed before the punctuation; for example [1], [1–3] or [1,3]. We have understood from this sentence that the references should be placed at the end of the sentence. We have moved the reference number in [] at the end of each individual sentence with Authors et. al.
- In the treatment sections, I suggest reporting an interesting new study about tirbanibulin:
- Campione, E.; Rivieccio, A.; Gaeta Shumak, R.; Costanza, G.; Cosio, T.; Lambiase, S.; Garofalo, V.; Artosi, F.; Lozzi, F.; Freni, C.; et al. Preliminary Evidence of Efficacy, Safety, and Treatment Satisfaction with Tirbanibulin 1% Ointment: A Clinical Perspective on Actinic Keratoses. Pharmaceuticals 2023, 16, 1686. https://doi.org/10.3390/ph16121686
Thank you for your suggestion. Indeed, the results of the study are very interesting and we have introduced a paragraph citing this paper.
- In the introduction section, specify that AK is a complex and multifactorial disease. I suggest reading and citing:
- Prezioso, C., Brazzini, G., Passerini, S., Di Fabio, C., Cosio, T., Bernardini, S., Campione, E., Moens, U., Pietropaolo, V., & Ciotti, M. (2022). Prevalence of MCPyV, HPyV6, HPyV7 and TSPyV in Actinic Keratosis Biopsy Specimens. Viruses, 14(2), 427. https://doi.org/10.3390/v14020427
Thank you for your suggestion. We will include the recommended article.
- The study design and methodology used for the research are unclear. Why did you not include Near-infrared spectroscopy? The material and method section is missing.
Thank you for your comment. Indeed, the methodology section is missing. We have completed the introductory part with the section Background and Motivation. We did not include a paragraph about near infrared spectroscopy because neither of the authors have practical experience with this method. We have introduced in this review only the imaging methods that are used by the authors in their current practice in order to have an expertise in the selection of included articles.
- Retinoids are not reported in the article. Explain.
Thank you for your observation, we have added a paragraph concerning the use of retinoids in AK treatment.
- Table 5 format font.
Thank you for your observation, we have made the corrections.
- In vitro, in vivoitalicized, please.
Thank you for this suggestion, we have made the changes accordingly.
We acknowledge #Reviewer 2 for the pertinent observations.
Reviewer 3 Report
Comments and Suggestions for Authors
Dear authors,
I think your topic is well contestualized. You describe very well AK and its possible progression to SCC.
I have only a minor revision to purpose for publish this manuscript.
Line 127: Add "in the next chapter" at the end of the phrase.
Tables and figures are complete and representative of topic.
References are appropriated and regards international literature.
Thank you
Author Response
Dear authors,
I think your topic is well contestualized. You describe very well AK and its possible progression to SCC.
I have only a minor revision to purpose for publish this manuscript.
Line 127: Add "in the next chapter" at the end of the phrase.
Tables and figures are complete and representative of topic.
References are appropriated and regards international literature.
Thank you.
We acknowledge #Reviewer 3 for the comments that definitely helped us to improve the manuscript.
Thank you for this observation. We have completed the phrase in order to be more specific.